# Impact of routine Newcastle disease vaccination on chicken flock size in smallholder farms in western Kenya

Elkanah Otiang[1,2,3], Samuel M. Thumbi[1,3,4], Zoë A. Campbell[5], Lucy W. Njagi[1], Philip N. Nyaga[1], Guy H. Palmer[3,4,6]*

1 College of Agriculture and Veterinary Sciences, University of Nairobi, Nairobi, Kenya, 2 Center for Global Health Research, Kenya Medical Research Institute, Kisumu, Kenya, 3 Washington State University Global Health-Kenya, Nairobi, Kenya, 4 Paul G. Allen School for Global Health, Washington State University, Pullman, Washington, United States of America, 5 International Livestock Research Institute, Nairobi, Kenya, 6 College of Health Sciences, University of Nairobi, Nairobi, Kenya

* gpalmer@wsu.edu

## Abstract

### Background

Poultry represent a widely held economic, nutritional, and sociocultural asset in rural communities worldwide. In a recent longitudinal study in western Kenya, the reported mean number of chickens per household was 10, with increases in flock size constrained principally by mortality. Newcastle disease virus is a major cause of chicken mortality globally and hypothesized to be responsible for a large part of mortality in smallholder flocks. Our goal was to determine the impact of routine Newcastle disease virus (NDV) vaccination on flock size and use this data to guide programs to improve small flock productivity.

### Methods

We conducted a factorial randomized controlled trial in 537 households: in 254 households all chickens were vaccinated every 3 months with I-2 NDV vaccine while chickens in 283 households served as unvaccinated controls. In both arms of the trial, all chickens were treated with endo- and ecto parasiticides every 3 months. Data on household chicken numbers and reported gains and losses were collected monthly for 18 months.

### Results

Consistent with prior studies, the overall flock size was small but with increases in both arms of the study over time. The mean number of chickens owned at monthly census was 13.06 ±0.29 in the vaccinated households versus 12.06±0.20 in the control households (p = 0.0026) with significant gains in number of chicks (p = 0.06), growers (p = 0.09), and adults (p = 0.03) in the vaccinated flocks versus the controls. Household reported gains were 4.50 ±0.12 total chickens per month when vaccinated versus 4.15±0.11 in the non-vaccinated controls (p = 0.03). Gains were balanced by voluntary decreases, reflecting household decision-making for sales or household consumption, which were marginally higher, but not

**Data Availability Statement:** Data can be found at https://osf.io/yn7fk/ (DOI: 10.17605/OSF.IO/YN7FK).

**Funding:** The study was supported by a gift of Paul G. Allen to Washington State University. The

funders had no role in study design, data collection and analysis, decision to publish, or preparation of the manuscript.

**Competing interests:** The authors have declared that no competing interests exist.

statistically significant, in vaccinated households and by involuntary losses, including mortality and loss due to predation, which were marginally higher in control households.

## Conclusion

Quarterly NDV vaccination and parasiticidal treatment resulted in an increase in flock size by a mean of one bird per household as compared to households where the flock received only parasiticidal treatment. While results suggest that the preventable fraction of mortality attributable to Newcastle disease is comparatively small relatively to all-cause mortality in smallholder households, there was a significant benefit to vaccination in terms of flock size. Comparison with previous flock sizes in the study households indicate a more significant benefit from the combined vaccination and parasiticidal treatment, supporting a comprehensive approach to improving flock health and improving household benefits of production in the smallholder setting.

## Introduction

Poverty and undernutrition, including both wasting and stunting, are major global challenges as illuminated in their importance to achieving the United Nations' Sustainable Development Goals [1]. Rural households in Africa, Asia, and Latin America disproportionately suffer from poverty and malnutrition [2, 3]. The Food and Agriculture Organization estimates that there are 500 million smallholder farms worldwide, which provide all or part of the household welfare for many of the 9% of the global population that living on less than US$2 per day [4–6]. Smallholder farms typically generate food and income from mixed crop agriculture, variably combined with small scale livestock production [7]. Chickens are the most commonly held livestock resource of smallholder households and represent an opportunity to provide eggs and meat to the household and potentially generate income from local sale [8, 9]. Furthermore, household chicken flocks represent an important economic and nutritional asset most commonly managed by women in rural households and can reflect their priorities for familial well-being [7, 8, 10].

Despite the global representation of chickens as the primary smallholder animal asset, small flock sizes and overall low productivity limit maximizing potential benefits [8, 11, 12]. Importantly, the low flock sizes do not appear to represent an economic based decision to optimize labour input while maximizing gain from household consumption or sale but rather a high level of involuntary losses due to mortality and predation [13]. A recent 4-year longitudinal study of 1,908 households in western Kenya found that the mean flock size was approximately 10 and highly stable over time, reflecting a balance of new chicks hatched on premises and losses, 60% of which were due to mortality [13]. Inputs into smallholder flocks were minimal: 98% of households reported that chickens scavenge for all or most of their feed during the day and 93% house chickens within the family dwelling at night [13]. Vaccination, supplemental nutrition, and treatment of endo- and ecto-parasites that would be expected to reduce morbidity and mortality [14] were uncommon [13].

Newcastle disease virus (NDV) is a highly transmissible and globally distributed infection of poultry [15, 16]. While high levels of biosecurity combined with vaccination are commonly used to prevent NDV outbreaks in commercial poultry, the reliance on free range scavenging for chickens in smallholder households results in ease of transmission between and within

flocks [17]. Consequently, NDV is widely considered to be the leading constraint to efficient smallholder poultry productivity in Africa [8]. While vaccination is highly effective under controlled conditions, its efficacy under smallholder conditions may be much more variable depending on the underlying health and age composition of the flock [9, 12, 16, 17].

To determine whether regularly scheduled NDV vaccination resulted in increases in flock sizes over time, we conducted a factorial randomized controlled trial in 537 households where all chickens in 254 households were vaccinated every 3 months with I-2 NDV vaccine while chickens in 283 households served as unvaccinated controls. Here we report the change in monthly flock census over an 18 months period and discuss the results in the context of improving flock productivity and household well-being.

## Materials and methods

### Ethical approval

The study was approved by the Ethical and Animal Care and Use Committees (SSC Protocol no. 3159) of the Kenya Medical Research Institute.

### Study population

The study took take place in Rarieda Sub-county of Siaya County in western Kenya within a health and demographic surveillance system (HDSS) site run by the Kenya Medical Research Institute and the United States Centres for Disease Control and Prevention (CDC) [18]. Livestock ownership is common in the area's households at 93%; 95% of households with an average flock size of ten chickens with chicken mortality accounting for over half of all reported animal death cases in participating households [13, 19].

### Study design

The field study was an 18-month factorial randomized controlled trial with 537 households enrolled and followed upon meeting an inclusion criteria of chicken ownership and grouped into two arms (vaccinated n = 254 and control n = 283). The sample size calculation used the assumption that Newcastle disease vaccination would decrease flock mortality by ≥10%, assuming a probability of type 1 error set at 0.05 and 80% detection power and further assuming 10% participant loss (household loss from study). This was based on average flock size per household of 10 chickens (range 4–60), accounting for unequal cluster sizes. The vaccinated group routinely received immunization of two drops of Newcastle disease virus (NDV) AVIVAX I-2 thermostable vaccine ($10^{9.7}$ egg infectious doses/ml) intranasally or intraocularly depending on chicken's age at recruitment and every three months thereafter. All vaccines were diluted by an animal health technician on the morning of the vaccination, maintained in a cool box at 4°C, and delivered prior to noon. Chickens in the control arm were not vaccinated. The same animal health technician dusted ecto-parasiticides of the group carbaryl (Sevin® powder) on the chickens and administered oral deworming using piperazine citrate (Ascarex-D®) in drinking water at recruitment and then every three months for chickens in both arms of the study.

### Data collection and analysis

Flock size and age composition of the flock was enumerated at each monthly visit by an animal health assistant blinded to the treatment groups. In addition, the individual responsible for flock or the household head was interviewed in the local language using a semi-structured questionnaire to collect recall data on increases and decreases to the flock during the prior three months. The data were collected using a mobile phone based application CommCare®

and data maintained through a Microsoft Access database®. The data were cleaned and analysed using STATA (Stata, 2013). The full data set and data dictionary are provided in Open Science Framework https://osf.io/yn7fk/

## Results

### Study population

A total of 537 rural households in Rarieda Sub-county of Siaya County in western Kenya were enrolled and followed longitudinally over 18 monthly visits starting in December, 2016. By random household distribution, 254 were grouped into the vaccination arm while 283 served as controls. The primary respondents to the questionnaire were most often the individuals who managed the chickens (89%) or the head of the compound (11%). Of the individuals responsible for management of the flock, 93% were women.

### Longitudinal monthly flock census

The mean flock sizes on visit 1, at the time of the first vaccination but prior to any possible effects of vaccination, were 11.63±0.70 for the 254 households in the vaccination arm of the study and 11.13±0.67 for the 283 control households. Enrolled households maintained flocks throughout the study period with less than 2% of visits recording no chickens at the monthly census. Over 18 monthly visits, the flock sizes increased in both arms but the total flock sizes were significantly greater in the vaccinated households: there was a cumulative mean of 13.06 ±0.29 chickens in the vaccinated households versus 12.06±0.20 in the control households (p = 0.0026) (Fig 1). The increases occurred across all age categories (Fig 1): the mean number of chicks in vaccinated households was 6.59±0.20 as compared to 6.20±0.13 in controls (p = 0.06), mean number of growers was 3.84±0.08 versus to 3.63±0.09 (p = 0.09), and mean number of adults was 3.32±0.19 as compared to 2.93±0.05 versus (p = 0.03). The increase was sustained throughout the study, whether analysed by the best-fit over the 18 visits (Fig 2) or by a best-fit tethered to the flock size at the visit 1 and then a best-fit determined (Fig 3).

### Household reported gains and losses

During each monthly visit, the household respondent was asked to self-report gains and losses during the prior month. Households that received vaccination reported gains of 4.50±0.12

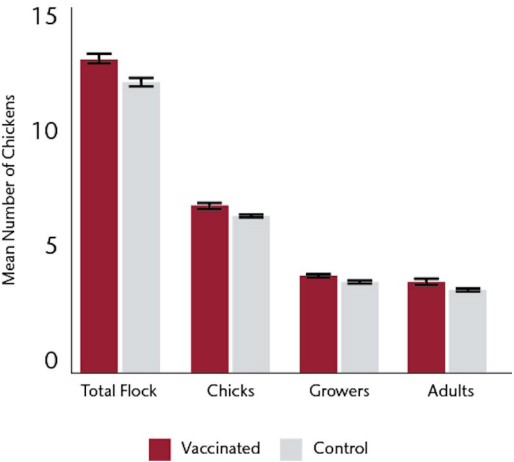

**Fig 1. Mean flock size at monthly census over 18 months.**

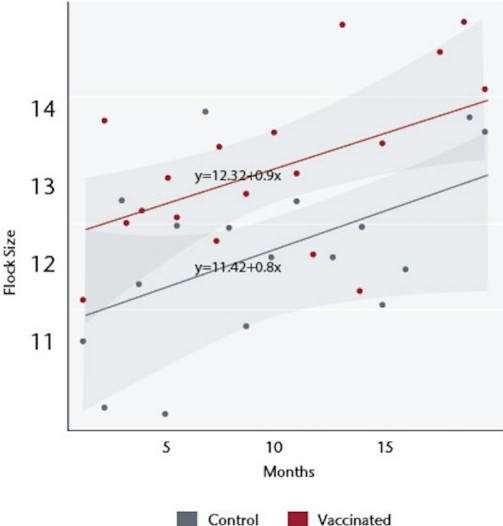

**Fig 2. Flock size dynamic over time (best-fit of all data points).**

chickens per month as compared to 4.15±0.11 in the non-vaccinated control households (p = 0.03). Vaccination households reported total decreases of 2.50±0.09 chickens per month versus 2.43±0.09 in the control households (p = 0.56). Reported voluntary decreases in flock size, reflecting household decision-making for sales or household consumption, were marginally greater in vaccinated households, 1.10±0.05, as compared to 1.03±0.04 in control households (p = 0.19), representing 44% and 42% of the monthly decreases in vaccinated and control households, respectively (Fig 4). Involuntary losses, including mortality and both unspecified loss and loss to predation, were reported to be marginally higher in control households, 1.4±0.08, as compared to 1.3±0.08 in vaccinated households (p = 0.39), with mortality representing the greatest reported source of loss in both groups (Fig 4).

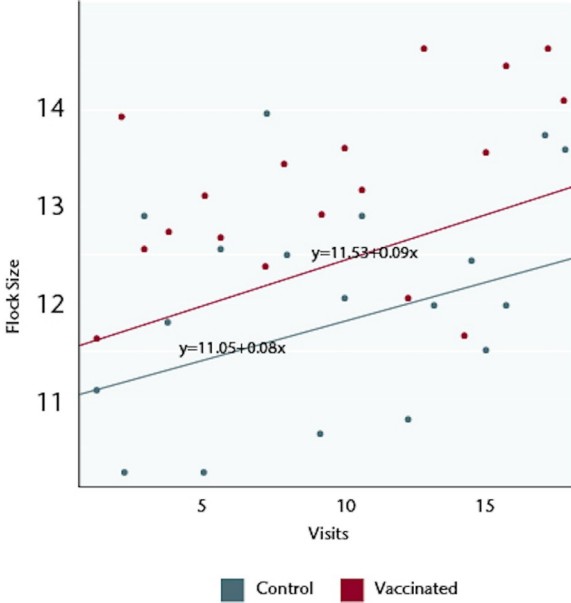

**Fig 3. Flock size dynamic over time (best-fit of data points tethered to initial flock size).**

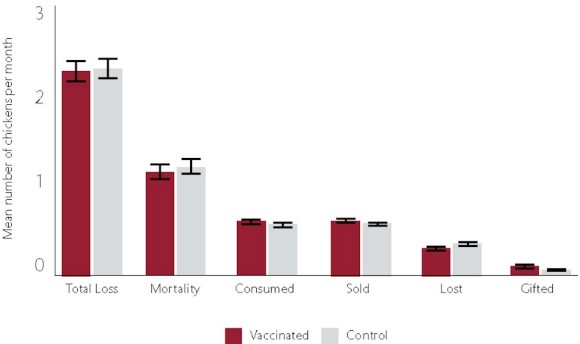

**Fig 4. Source of reported monthly off-take in chickens.**

## Discussion

Newcastle disease is endemic at the village level in Kenya and specifically in this region of western Kenya [20], where greater than 50% of chicken mortality is attributed to NDV. The use of a relatively large factorial randomized controlled trial allowed determination of the impact of NDV vaccination on flock size. Quarterly NDV vaccination of all chickens resulted in a mean gain of one chicken per household. While this gain seems modest, it represents an 8–10% increase in flock size from vaccination against a single viral pathogen. Importantly, the increase was sustained throughout the 18-month study, which included both wet and dry seasons to account for seasonal variation in transmission [20]. This suggests the opportunity for increased gains over time and an accumulating impact of routine NDV vaccination. If these gains were utilized for either household nutrition or income from sale there would be measurable benefits to the family [21, 22]. Consumption of either eggs or chicken meat have been shown to reduce childhood stunting [22], which remains at a high level in the study region and throughout much of rural sub-Saharan Africa. Two prior studies in this population support the impact of larger flock size on nutritional gains. The first, a study of 1500 households, found that an increase in number of chickens per household associated with a 28% likelihood of childhood consumption of eggs in the prior 3 days, holding other household factors constant [19]. The second study of 1800 households showed that poultry ownership was linked to a significant increase in both egg and chicken consumption (adjusted incidence rate ratio of 1.3). Furthermore, consumption was associated with a significant increase in monthly child height gain for children over the age of 6 months [22]. Similarly, routine vaccination has potential to increase household income. In a recent study of smallholder households in rural Tanzania, the market price for vaccine to inoculate 10 chickens was US$1.20 while the local market price for an adult chicken was US$3.12 [23]. Notably, respondents in that study were willing to pay twice the market price for vaccine, reflecting that households valued vaccination and perceived a favorable return on investment [23]. As 93% of the individuals in this study that managed the flocks were women and women have been shown to devote a much greater proportion of income into family nutrition and health care, this relatively modest increase in income can have a significant impact on familial well-being [7, 10, 11].

The current study allowed an estimation of the preventable fraction of mortality due to NDV. Vaccination was carried out by qualified animal health technicians supervised by a licensed veterinarian and records were kept on the storage and delivery of the I-2 vaccines. While NDV I-2 vaccination has been shown in numerous experimental and field studies to be highly effective [24–26], the preventable fraction of mortality due to a single vaccine reflects the overall causes of mortality and varies depending on the specifics of poultry management at

the household level [17, 23]. The most reliable measure of vaccine impact from this study would be the 8–10% increase in total flock size at monthly census. In contrast to commercial poultry vaccination programs, in this study chickens were not uniformly vaccinated at a given age but on a quarterly schedule when all chickens on the premises were vaccinated regardless of prior vaccination history. This variation in vaccination history and immune status of individual birds affects the level of population immunity and would be likely to significantly diminish the flock level efficacy of vaccination. However, this variation in age structure for vaccination is reflective of smallholder household flock management.

The self-reported mortality by the household respondents indicated no significant difference in mortality between the vaccinated and control flocks (Fig 4). This discrepancy between independent census data and self-reported data has been previously observed among households in the study region [13]. In the prior study, individuals consistently overestimated gains and underestimated losses relative to actual census data [13]. This pattern is observed in the current study: as an example, the self-reported total monthly gains in vaccinated households were reported as 4.5 with overall monthly decreases of 2.5, inconsistent with census data that indicates a smaller monthly increase. Recall bias, representing systemic errors in remembering past events, and social desirability bias are two possible explanations for the discrepancy [27]. Households in the study region do not maintain written records on flock size, gains, and losses, thus losses that occurred earlier in the month may be discounted relative to gains that are still represented in the flock. This may be especially true for young chicks, which have a high mortality rate from NDV as well as other infectious and non-infectious causes [13, 15, 28, 29]. Social desirability bias, the tendency for survey respondents to answer questions in a way that will be viewed favourably by others, may also have an impact as the household may want to be seen by the interviewers and animal health team as being a responsible member of the community and thus overstate gains and understate involuntary losses, including mortality [30].

Notably, there were sustained gains in flock size in households that received vaccination and parasiticidal treatment and the control households that received only parasite control. While a control group with no treatment was not included (as participation required time commitment by the respondents), comparison with both the flock sizes at enrolment and the historical mean flock size of 10 in this study site [13, 19] suggest that parasiticidal treatment had a significant effect alone, which was further enhanced by NDV vaccination. Furthermore, quarterly visitation by an animal health technician provided the opportunity to seek *ad hoc* advice on flock management. This opportunity and the impact of external interest in a household's flock size may also have improved management independent of or interacting with vaccination and parasiticidal treatment. This is consistent with prior studies showing the impact of combined interventions and emphasizes that a comprehensive approach to improved poultry management at the smallholder level is needed [8]. Integrating supplemental nutrition would highly likely increase the efficacy of vaccination as well as maximize benefits from parasite control. The lack of routine vaccination for smallholder flocks does not appear to reflect a lack of knowledge regarding the importance of vaccination as indicated by willingness to pay studies in which respondents were willing to pay more than the actual cost of NDV vaccine [23]. Rather, the primary barrier to improved management appears to be at the level of services delivery. At a household level the incentives for effective delivery of more comprehensive poultry health and husbandry services are too small for commercial investment. However, at a community level this may provide a larger integrated market that would attract commercial engagement, especially if incentivized by government support for rural communities.

Finally, whether increased flock sizes are desirable from a labor management perspective is important, especially given that women, who most commonly have primary responsibility for

flock husbandry management in this region, have multiple other demands on their effort. In a prior study [13], we assessed whether poultry owning households in this region maintained relatively small flock sizes as a deliberate decision to maximize benefits per unit labor by voluntary reduction of chicken numbers by consumption or sale versus involuntary losses due to mortality, predation or theft. The overwhelming majority of off-take was involuntary, principally due to mortality, that does not reflect the owner's decision to maximize value through nutritional gain, income, or social capital. This strongly suggests that there is substantial opportunity to enhance the value of chickens as an asset, both nutritional and income generating, for smallholder households.

## Conclusion

This study demonstrates a significant impact of NDV vaccination on overall flock size that is maintained over time and is enhanced by parasite control. This is consistent with the need for integrated control of infectious diseases of poultry with substantial opportunity to improve nutritional and economic security for rural smallholder households.

## Acknowledgments

We thank the contributions of the following individuals, who made the research possible. Field team supervisor, James Oigo; Animal Health Technicians, Samwel Asembo, Geoffrey Odima, Bob Miseda, Beryl Oyoo and Austin Ochung'; Community Health Interviewers: Joseph Onyango, Daniel Odongo, Rosemary Warinda, Bridon Kojo, Thomas Wachiaya, Fredrick Ong'wen, Meresia Owuor; Martin Aundi, Fredrick Adhiambo, Benedine Warinda, Jane Okal, Fred Ojode; Data Programmer and Manager, Linus Otieno; Data Clerk, Judith Oduol.

## Author Contributions

**Conceptualization:** Zoë A. Campbell, Guy H. Palmer.

**Data curation:** Elkanah Otiang.

**Formal analysis:** Elkanah Otiang, Samuel M. Thumbi, Zoë A. Campbell, Guy H. Palmer.

**Investigation:** Elkanah Otiang, Samuel M. Thumbi.

**Methodology:** Guy H. Palmer.

**Project administration:** Guy H. Palmer.

**Resources:** Guy H. Palmer.

**Supervision:** Samuel M. Thumbi, Lucy W. Njagi, Philip N. Nyaga.

**Visualization:** Guy H. Palmer.

**Writing – original draft:** Elkanah Otiang, Guy H. Palmer.

**Writing – review & editing:** Elkanah Otiang, Samuel M. Thumbi, Zoë A. Campbell, Lucy W. Njagi, Philip N. Nyaga, Guy H. Palmer.

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
