## [Decision Letter · Decision Letter 0]

17 Feb 2021

PONE-D-21-01892

Marginal impact of routine Newcastle Disease vaccination on chicken flock size in smallholder farms in western Kenya

PLOS ONE

Dear Dr. Palmer,

Thank you for submitting your manuscript to PLOS ONE. After careful consideration, we feel that it has merit but does not fully meet PLOS ONE’s publication criteria as it currently stands. Therefore, we invite you to submit a revised version of the manuscript that addresses the points raised during the review process.

After careful review, two reviewers have provided favourable reviews of this manuscript. I am returning to the authors with the recommendation of minor revisions to encourage integration of the few changes suggested by the reviewers, at which point this will be appropriate for acceptance.

We look forward to receiving your revised manuscript.

Kind regards,

Eric Fèvre

Academic Editor

PLOS ONE

Additional Editor Comments (if provided):

After careful review, two reviewers have provided favourable reviews of this manuscript. I am returning to the authors with the recommendation of minor revisions to encourage integration of the few changes suggested by the reviewers, at which point this will be appropriate for acceptance.

Journal Requirements:

Reviewers' comments:

Reviewer's Responses to Questions

**Comments to the Author**

1. Is the manuscript technically sound, and do the data support the conclusions?

Reviewer #1: Yes

Reviewer #2: Yes

2. Has the statistical analysis been performed appropriately and rigorously? 

Reviewer #1: Yes

Reviewer #2: Yes

3. Have the authors made all data underlying the findings in their manuscript fully available?

Reviewer #1: Yes

Reviewer #2: Yes

4. Is the manuscript presented in an intelligible fashion and written in standard English?

Reviewer #1: Yes

Reviewer #2: Yes

5. Review Comments to the Author

Reviewer #1: PONE-D-21-01892 Marginal impact of routine Newcastle Disease vaccination on chicken flock size in smallholder farms in western Kenya by Otiang, Thumbi, Campbell, Njagi, Nyaga, and Palmer describe the study they performed to ascertain the impact of Newcastle disease virus vaccination on the flock size in smallholder flocks in western Kenya. Original data are presented and don’t appear to have been previously published.

There is no need to capitalize the “d” in the word disease in the title and throughout the manuscript: Newcastle disease virus or Newcastle disease.

As you stated in the discussion, “this study demonstrates a significant impact of NDV vaccination on overall flock size that is maintained over time and is enhanced by parasite control.” Thus, I disagree with the word marginal in the title. The two sentences seem to be opposite. Perhaps you can remove the word marginal: Impact of routine Newcastle disease vaccination on chicken flock size in smallholder farms in western Kenya. That way you get to explain in the paper how the marginal difference in bird numbers in the two groups actually translates to an actual impact in nutrition and finances of the family.

In the materials and methods section is not clear who did the vaccination every three months and how this relates to the survey collecting data every three months. Was the person who provided the survey also providing the vaccine (to ensure it was kept at 4C)? If not, where did the person obtain the vaccine and how long after receiving it did they have to catch and vaccinate 13+ chickens? I now see some of this info. in the discussion (line 201)—perhaps include some of it in the materials and methods. Also include who dilutes the vaccine, when the vaccine is diluted in relation to when it administered and the intended target vaccine titer given to each bird. This is for the purposes of allowing others who would like to replicate your study the parameters that you used.

What was the average number of vaccines that each chicken received? Chickens would have to have at least two vaccines to be considered protected depending if any maternal antibodies were present to neutralize the live vaccine administered. If there is a constant turnover vaccination might be less effective.

Did all of the smallholders keep chickens each month or were there months they did not have any chickens?

Line 132 data is plural. The data were collected…

There is no description of any birds getting sick and/or dying of respiratory disease in a country where virulent Newcastle disease virus is endemic. There are no questions about possible AIV or NDV respiratory deaths.

Many have published that NDV occurs with seasonality for some countries in Africa. Of the 26 dates with the highest amounts of total mortality 12.7% occur in Dec, Jan and Feb and almost 7% in April and May. Does this correspond to a dry or wet season?

Also, birds with partial protection often have neurological signs, potentially without mortality. What is the usual outcome when an owner observes neurological or respiratory symptoms? Are the birds consumed?

Reviewer #2: This is a well structured, well presented article describing a longitudinal study of the impact of ND vaccine on flock size in smallholder households in Western Kenya.

The authors do well to lay out the context of smallholder poultry in livelihoods in low and middle income countries, before getting into the meat of the study.

It is extremely difficult to measure flock/herd size changes, and the study is well structured in terms of sample sizes, time of study, and different methodologies built in to measure such changes.

A few queries:

Are there any other factors which might account for these changes, other than ND vaccine? The presence of veterinary assistants who apply the vaccine may also be consulted for advice of other issues while vaccinating, can the mere attendance of a veterinary assistant administering technical interventions be ruled out as having any contributory effect? Could any other factors account for the differences seen?

What flock size is manageable by the different smallholders in this region? Is the increase in flock size always a good thing? With more poultry running around, is there a greater likelihood of predators or thefts, which would discourage flock size beyond a manageable limit?

In the conclusions there are some huge assumptions that flock size changes will result in better household nutrition and income, which are not supported by data specific to this population. And there is also an assumption of what women would do with the flock size changes, which might well be true, but are not supported by data from this study.

6. PLOS authors have the option to publish the peer review history of their article (what does this mean?). If published, this will include your full peer review and any attached files.

Reviewer #1: No

Reviewer #2: No

---

## [Author Response · Author response to Decision Letter 0]

27 Feb 2021

We appreciate the reviewers’ time and effort in the critique of the manuscript and have responded to the comments below and note the specific changes in the revised manuscript.

Reviewer 1: The reviewer had nine specific suggestions, each of which is addressed below and with changes to the manuscript.

1.1 “There is no need to capitalize the “d” in the word disease in the title and throughout the manuscript”. We have made this change throughout the manuscript.

1.2 “As you stated in the discussion, ‘this study demonstrates a significant impact of NDV vaccination on overall flock size that is maintained over time and is enhanced by parasite control.’ Thus, I disagree with the word marginal in the title”. We have revised the title accordingly.

 1.3 “In the materials and methods section is not clear who did the vaccination every three months and how this relates to the survey collecting data every three months. Was the person who provided the survey also providing the vaccine (to ensure it was kept at 4C)? If not, where did the person obtain the vaccine and how long after receiving it did they have to catch and vaccinate 13+ chickens? I now see some of this info. in the discussion (line 201)—perhaps include some of it in the materials and methods. Also include who dilutes the vaccine, when the vaccine is diluted in relation to when it administered and the intended target vaccine titer given to each bird. This is for the purposes of allowing others who would like to replicate your study the parameters that you used.” Households were contacted the day before the vaccination/data collection and requested to keep their chickens housed the following day (often this is within the home). The vaccinator, a licensed animal health technician, diluted vaccines in the morning and maintained them in a 4oC cold box until delivered to the birds. The vaccine vial contains 109.7 EID50/ml with a delivered target of 108.7 EID50 per bird. All vaccinations were delivered before noon each day. The same individual was also responsible for parasiticidal treatment for both groups. Data was collected monthly by enumerators who were blinded to the treatment groups. This has been added to the Materials and methods.

1.4 “What was the average number of vaccines that each chicken received? Chickens would have to have at least two vaccines to be considered protected depending if any maternal antibodies were present to neutralize the live vaccine administered. If there is a constant turnover vaccination might be less effective.” All birds on the premises were vaccinated at each 3 month interval, regardless of prior vaccination status. This does introduce some variation in terms of immunity in individual birds and thus at the population basis. While this is reflective of household chicken populations and recommended vaccination practices, it may affect vaccine efficacy. We have added this to the Discussion.

1.5 “Did all of the smallholders keep chickens each month or were there months they did not have any chickens?” All households had chickens at enrollment. During the study, 111 households reported no birds at a monthly census (<2% of observations). This had been added to the Results.

1.6 “Line 132 data is plural. The data were collected…” This has been corrected.

1.7 “There is no description of any birds getting sick and/or dying of respiratory disease in a country where virulent Newcastle disease virus is endemic. There are no questions about possible AIV or NDV respiratory deaths.” In our prior study of 1,908 poultry owning households (which overlap with those of the present study) 66% of deaths occurred in birds showing nervous signs and 58% with respiratory signs, both consistent with diagnosis of Newcastle disease. Based on presumptive diagnoses by animal health technicians, Newcastle disease was responsible for 52% of mortality followed by Fowl Pox and Infectious Bursal Disease. Although not confirmed by laboratory diagnosis, this is consistent with a high incidence of Newcastle disease in these households (Otiang E et al., Mortality as the primary constraint to enhancing nutritional and financial gains from poultry: A multi-year longitudinal study of smallholder farmers in western Kenya. PLoS One 15(5):e0233691, 2020). We have revised the Discussion to reference the incidence of Newcastle disease in this region. 

1.8 “Many have published that NDV occurs with seasonality for some countries in Africa. Of the 26 dates with the highest amounts of total mortality 12.7% occur in Dec, Jan and Feb and almost 7% in April and May. Does this correspond to a dry or wet season?” The December-February period bridges the transition from the “short rains” wet season in December to the dry months of January and February. April and May are definitely in the “long rains” wet season. We have added this to the Discussion along with a relevant review that summarizes seasonal data in Kenya (Apopo AA et al., A retrospective study of Newcastle disease in Kenya. Tropical Animal Health and Production 52:699-710, 2020).

 1.9 “Also, birds with partial protection often have neurological signs, potentially without mortality. What is the usual outcome when an owner observes neurological or respiratory symptoms? Are the birds consumed?” Households report that sick birds are normally buried or burned with only 10% reporting subsequent consumption.

Reviewer 2: The reviewer had three specific suggestions, each of which is addressed below and with changes to the manuscript.

2.1 “Are there any other factors which might account for these changes, other than ND vaccine? The presence of veterinary assistants who apply the vaccine may also be consulted for advice of other issues while vaccinating, can the mere attendance of a veterinary assistant administering technical interventions be ruled out as having any contributory effect? Could any other factors account for the differences seen?” The reviewer’s point is supported by the sustained increases in poultry numbers in households where chickens were either vaccinated or unvaccinated households. In addition to the provision of endo- and ecto-parasiticidal treatments to both groups, the presence of the veterinary assistants in both groups may contribute to the increase, due either to specific advice on husbandry or the response of the caretaker to external interest in flock size. We have added this to the revised Discussion.

2.2 “What flock size is manageable by the different smallholders in this region? Is the increase in flock size always a good thing? With more poultry running around, is there a greater likelihood of predators or thefts, which would discourage flock size beyond a manageable limit?” This is a highly relevant question. In a prior study, we assessed whether poultry owning households in this region maintained these relatively small flock sizes as a deliberate decision to maximize benefits per unit labor by voluntary reduction of chicken numbers by consumption or sale versus involuntary losses due to mortality, predation or theft (Otiang E et al., Mortality as the primary constraint to enhancing nutritional and financial gains from poultry: A multi-year longitudinal study of smallholder farmers in western Kenya. PLoS One 15(5):e0233691, 2020). The overwhelming majority of off-take can be classified as involuntary off-take, principally due to mortality, that does not reflect the owner’s decision to maximize value through nutritional gain, income, or social capital. This strongly suggests that there is substantial opportunity to enhance the value of chickens as an asset, both nutritional and income generating, for smallholder households living at poverty level. We have emphasized this in the revised manuscript.

2.3 “In the conclusions there are some huge assumptions that flock size changes will result in better household nutrition and income, which are not supported by data specific to this population. And there is also an assumption of what women would do with the flock size changes, which might well be true, but are not supported by data from this study.” We appreciate this point. Two prior studies in this population support the impact of larger flock size on nutritional gains. The first, a study of 1500 households, found that an increase in number of chickens per household associated with a 28% likelihood of childhood consumption of eggs in the prior 3 days, holding other household factors constant (p<0.001) (Thumbi SM et al., Linking human health and livestock health: a "one-health" platform for integrated analysis of human health, livestock health, and economic welfare in livestock dependent communities. PLoS One 10:e0120761, 2015). The second study of 1800 households showed that poultry ownership was linked to a significant increase in both egg and chicken consumption (adjusted incidence rate ratio of 1.3). Furthermore, consumption was associated with a significant increase in monthly child height gain for children over the age of 6 months (Mosites EM et al., Child height gain is associated with consumption of animal-source foods in livestock-owning households in western Kenya. Public Health Nutrition 12:1-10, 2017). We have emphasized these finding, accompanied by the relevant references.

---

## [Editor Report · Decision Letter 1]

2 Mar 2021

Impact of routine Newcastle disease vaccination on chicken flock size in smallholder farms in western Kenya

PONE-D-21-01892R1

Dear Dr. Palmer,

We’re pleased to inform you that your manuscript has been judged scientifically suitable for publication and will be formally accepted for publication once it meets all outstanding technical requirements.  Please see the additional editorial note below.

Kind regards,

Eric Fèvre

Academic Editor

PLOS ONE

Additional Editor Comments (optional):

Thank you for the revisions and the care taken with your response. The manuscript is now acceptable for publication. Reading through the comments and the responses, I wonder whether, when discussing flock sizes and the change in farming systems in the region under study, it might be relevant to cite the following paper also: https://doi.org/10.1017/S175173112000110X, which discusses the trajectory of poultry intensification in this region.
---

## [Editor Report · Acceptance letter]

9 Mar 2021

PONE-D-21-01892R1 

Impact of routine Newcastle disease vaccination on chicken flock size in smallholder farms in western Kenya 

Dear Dr. Palmer:

I'm pleased to inform you that your manuscript has been deemed suitable for publication in PLOS ONE. Congratulations! Your manuscript is now with our production department. 

Kind regards, 

on behalf of

Prof. Eric Fèvre 

Academic Editor

PLOS ONE